# Sociocultural Norm Similarities and Differences via Situational Alignment and Explainable Textual Entailment

**Sky CH-Wang**°*    **Arkadiy Saakyan**°*    **Oliver Li**°    **Zhou Yu**°    **Smaranda Muresan**°•

°Department of Computer Science, Columbia University

•Data Science Institute, Columbia University

{skywang, a.saakyan}@cs.columbia.edu

## Abstract

Designing systems that can reason across cultures requires that they are grounded in the norms of the contexts in which they operate. However, current research on developing computational models of social norms has primarily focused on American society. Here, we propose a novel approach to discover and compare descriptive social norms across Chinese and American cultures. We demonstrate our approach by leveraging discussions on a Chinese Q&A platform—知乎 (Zhihu)—and the existing SOCIALCHEMISTRY dataset as proxies for contrasting cultural axes, align social situations cross-culturally, and extract social norms from texts using in-context learning. Embedding Chain-of-Thought prompting in a human-AI collaborative framework, we build a high-quality dataset of 3,069 social norms aligned with social situations across Chinese and American cultures alongside corresponding free-text explanations. To test the ability of models to reason about social norms across cultures, we introduce the task of explainable social norm entailment, showing that existing models under 3B parameters have significant room for improvement in both automatic and human evaluation. Further analysis of cross-cultural norm differences based on our dataset shows empirical alignment with the social orientations framework, revealing several situational and descriptive nuances in norms across these cultures.

## 1 Introduction

Social norms are normative beliefs that guide behavior in groups and societies (Sherif, 1936). Deviance from these expectations of behavior can cause perceptions of impoliteness (Culpeper, 2011), feelings of offense (Rubington and Weinberg, 2015), and pragmatic failure (Thomas, 1983). Social norms vary across cultures (Triandis et al., 1994; Finnemore, 1996), and different cultural

---

*Equal Contribution.

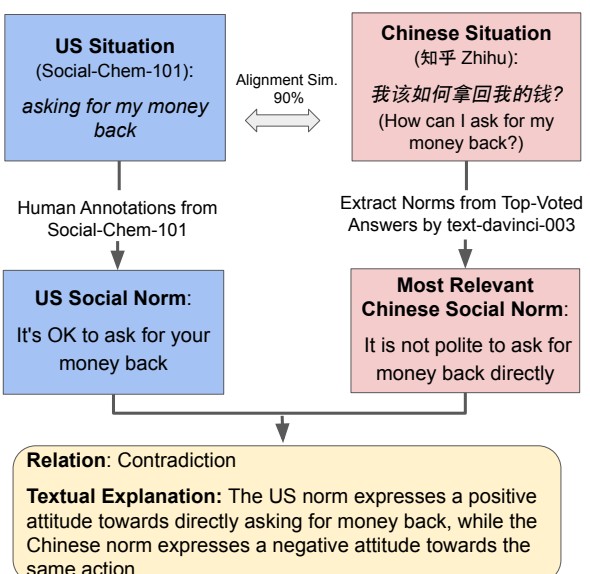

Figure 1: An example of descriptive norms conditioned on an aligned cross-cultural situation, together with their inference relation and corresponding textual explanation.

norms can lead to conflict within intercultural interactions due to perceptions of deviance (Durkheim, 1951). Creating computational systems that can robustly reason and translate across cultures in pragmatic communication requires that they be grounded in these norms and their differences across contexts. As an initial step in addressing this question, we propose a novel approach to discover and compare social norms conditioned on social situations across Chinese and American cultures. Leveraging 知乎 (Zhihu), a Chinese Q&A platform, alongside the existing SOCIALCHEMISTRY (Forbes et al., 2020) dataset on social norms as respective proxies of Chinese and American cultural axes, our paper offers the following contributions:

- **A human-AI collaboration framework for cross-cultural descriptive norm discovery** consisting of (1) automatic situation alignment using cross-lingual similarity between

SOCIALCHEMSTRY situations and questions from Zhihu, (2) Chinese social norm extraction from Zhihu answers using few-shot prompting with GPT-3 (Brown et al., 2020), (3) cross-cultural norm similarity and difference identification as textual entailment with explanations using GPT-3 with Chain of Thought (CoT) Prompting (Wei et al., 2022); and (4) human feedback in verification and editing. An example of outputs is shown in Figure 1.

- **A new dataset for cross-cultural norms understanding with explainable textual entailment**. Our human-AI collaboration enables us to create a novel dataset of 3069 situation-aligned entailment pairs of Chinese and American norms together with textual explanations. We introduce the new task of explainable social norm entailment and show that it is challenging for models fine-tuned on related tasks; fine-tuning on our task directly still leaves significant space for improvement.

- **An analysis of cross-cultural differences in social norms enabled by our dataset.** In Section 6, we show that the analysis enabled by our dataset empirically aligns with prior work on differences in Chinese and American cultures. Our empirical results align with the social orientations framework (Yang, 1993) in understanding Chinese-American norm differences and reveal several situational and descriptive nuances in norms across these cultures.

## 2 Related Work

Our work is situated in the broader literature on the study of social norms (Sherif, 1936) and how they vary across cultures (Thomas, 1983). Here, our work is rooted specifically in the study of descriptive norms (Rawls, 1951, 2004; Cialdini et al., 1990)—what people *actually* do, rather than prescriptive norms, or what they think people *ought* to do—and focuses on the differences between Chinese and American cultures.

We build on recent computational work in creating systems capable of situated reasoning in social situations, most closely adapting the rule-of-thumb formalism for descriptive norms introduced in Forbes et al. (2020). This line of work not only spans that of commonsense reasoning (Sap et al., 2019; Rashkin et al., 2018), but also in judg-ments of appropriate and ethical behavior (Emelin et al., 2021; Jiang et al., 2022) and in grounding behavior in areas like dialogue (Ziems et al., 2022) and situated question answering (Gu et al., 2022a) more specifically on underlying knowledge of social norms. In recognizing that social norms are often culturally (Haidt et al., 1993) and even demographically (Plepi et al., 2022; Wan et al., 2023) specific, prior work in this area has primarily revolved around the normative judgments of majority English-speaking cultures represented within North America. In contrast, here, aligning with the broader goal of creating systems that can effectively reason *across* cultures and languages (Liu et al., 2021), we focus on computationally studying norms across Chinese and American cultures, expanding on the utility of large language models in ways that have the potential to transform modern computational social science (Ziems et al., 2023).

Contemporary studies of Chinese cultural societies (Yang, 1993) emphasize several broad differences relative to American culture. Under the framework of social orientations, emphasis in Chinese society is placed especially on family, relationship, authority, and personal reputation social orientations. In particular, past work has shown a large significance compared to American culture in familistic collectivism and harmony (Ch'eng-K'Un, 1944; Yang, 1988; Campos et al., 2014), relational determinism (Chen and Chen, 2004; Chua et al., 2009), and authority worship (Yang, 1970; Thornton and Fricke, 1987; Hu, 2016), among other factors, in influencing social behavior. Critical reviews of past cross-cultural work have criticized weaknesses in study design and their overly broad generalizations (Voronov and Singer, 2002), in favor of a more fine-grained analysis. Here, under the framework of descriptive norms and sourcing data from social media at scale, we conduct a more nuanced analysis of how norms vary across cultures under situational controls.

## 3 Data Sources

Our analysis of social norm variation across Chinese and American contexts draws from the largest Q&A discussion platform in China—知乎 (Zhihu)—and existing data gathered by SOCIAL-CHEMISTRY (Forbes et al., 2020), treating these data sources as different cultural axes.

**Social Chemistry 101 (Forbes et al., 2020)** is a natural language corpus of ethical judgments

| |
|---|
| **Question**: 当有人说家里有丧事时应如何回应更礼貌? (How to respond politely when someone says there is a funeral in the family?)
**Answer**: 节哀。可以礼貌性的拍拍肩膀或者嘱咐对方虽然最近会比较操劳但也要注意身体。(Condolences. You can politely pat on their shoulders or tell them to pay attention to their health even though it will be hard for them recently.) |
| **RoT 1**: it is appropriate to say "节哀" to someone who has lost a family member or friend
**RoT 2**: it is appropriate to pat someone on the back to show your sympathy to them
**RoT 3**: it is appropriate to tell them to take care of themselves though they are sad |

Table 1: Example of a Zhihu question-answer pair (top) with English translations and relevant social norms in rules-of-thumb form (bottom).

and social norms on everyday situations. Crowd-sourced social norms on situations scraped from Reddit (i.e., `r/AmITheAsshole`) and other sources are expressed in free-text *rule-of-thumb* form, where each rule-of-thumb consists of a judgment on an action, i.e., "It's rude to run the blender at 5 AM". Annotators of this dataset are 55% women, and 94% of annotators had spent 10 or more years in the United States; as such, we treat the normative judgments represented within this dataset as originating from a predominantly English-speaking, American context[1].

知乎 **Zhihu**   has over 101 million monthly active users (Zhihu, 2022) and is one of China's largest online knowledge communities. Similar to its international counterparts like Quora, Zhihu is primarily a Q&A site where users can post questions, answer existing questions, and up-vote answers to questions by other users, among other platform interactions. Users are 59% women (Yang, 2022a) and are primarily educated and middle-class, with research showing that over 80% of its users possess at least a bachelor's degree (Zhang, 2020). This population is significant because the middle class is the primary upholder of social stability (Goodman, 2014) and the primary driver of cultural trends and public opinion (Denemark and Chubb, 2016) in China. An example of a Zhihu question-answer pair is shown in Table 1.

Questions on Zhihu are often posed as to in-

---

[1]Despite the many demographic similarities between our two cultural axes, a few notable differences remain. Most notably of which is class composition, where only 47% of annotators of Social Chemistry 101 considered themselves to be middle-class compared to over 80% of users on Zhihu. See our Limitations section for a discussion on potential impacts.

| |
|---|
| **Entailment Relation** |
| **Situation**: *I think I'm jealous of my best friend.*
**US Norm**: You should be proud of people and not jealous of them.
**Chinese Norm**: It is not appropriate to be jealous of your best friend.
**Textual Explanation**: Both norms express disapproval of being jealous of someone, especially a best friend. |
| **Contradiction Relation** |
| **Situation**: *telling my teacher a classmate cheated on a test.*
**US Norm**: It's wrong to be a tattle-tale.
**Chinese Norm**: It is not wrong to report a cheating student to the teacher $< \cdots >$
**Textual Explanation**: The US norm expresses disapproval of telling on a classmate, while the Chinese norm expresses neutrality towards the action. |

Table 2: Examples of similarities (entailments, top) and differences (contradictions, bottom) in situated norms between Chinese and American cultures from our dataset, alongside relation explanations.

quire about the appropriate action to partake in a given situation, i.e., "what do I do if my friend is cheating on their exam?", with other users indicating their views on appropriate courses of action in the answers below. Reasoning that these questions align most closely in form to situations in SOCIALCHEMISTRY, we translate and query for all situations present in the SOCIALCHEMISTRY dataset through Zhihu's search function and save the top 100 question posts returned for every translated SOCIALCHEMISTRY situation. Following the rationale that social norms are the most common and broadly accepted judgments of actions in given situations, we take the most up-voted answer[2] to each question as the basis from which we extract social norms, described in the following section. In total, we obtained answers to 508,681 unique questions posed on Zhihu. The study and data collection were approved by an Institutional Review Board prior to data collection; a detailed breakdown of this corpus is detailed in Appendix Section A. We reiterate both datasets can only be used as a *proxy* for social norms in each culture and further discuss the limitations of our approach and these assumptions in Section 7.

## 4   Human-AI Collaboration to Create a Cross-Cultural Social Norm Dataset

We enable our computational analysis of social norm variation across Chinese and American cultures through a framework of (1) automatic sit-

---

[2]See Limitations for a discussion on method caveats.

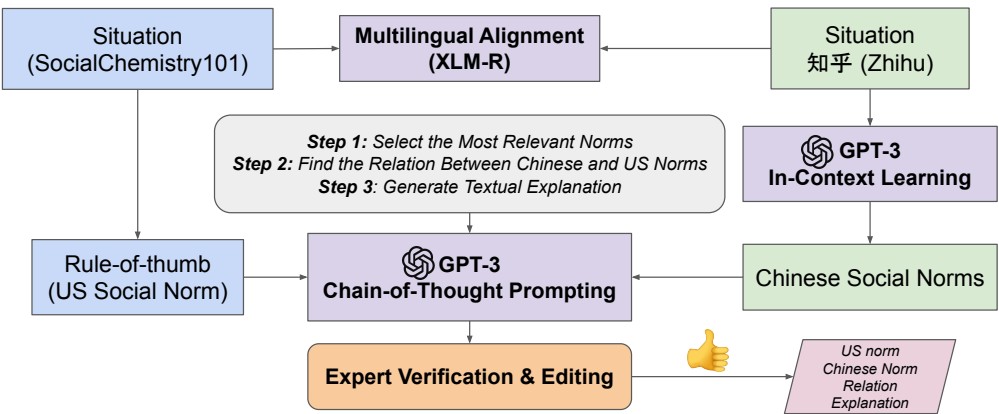

Figure 2: Our human-AI collaboration framework for creating a cross-cultural Chinese-American social norm NLI dataset through (1) situation alignment, aligning cross-lingual situations between Zhihu and Social Chemistry, (2) social norm extraction, from free-form answers to rules-of-thumb with in-context learning, and (3) norm relation inference with textual explanation with CoT prompting, coupled with (4) expert verification and editing.

uation alignment, (2) social norm extraction, (3) cross-cultural similarity/difference identification as textual entailment with explanations, and (4) human feedback. An illustration of our human-AI collaboration framework is presented in Figure 2.

**Aligning Situations Across Cultures.** Descriptive judgments of appropriate behavior are often situationally-dependent (Leung and Morris, 2015); an accurate comparison of norms across cultures must first ensure a similarity of contexts. For instance, Yamagishi et al. (2008) showed that Japanese preferences for conformity disappeared in private compared to when they were observed. To obtain the closest matching situations between US and Chinese contexts, we align situations found in the SOCIALCHEMISTRY dataset with similar questions on Zhihu. While the Zhihu search API allows for direct query searches of platform content, semantic alignment of returned results with queries remains low due to an observed bias in prioritizing entity matches. To improve this alignment of situations between US and Chinese contexts, we use XLM-R (Conneau et al., 2020) to obtain cross-lingual representations of situations and further perform one-to-one matching through cosine similarity. Setting the similarity threshold to 0.895 allowed us to align 3069 situations, of which around 80% were correctly aligned, based on a manual review of a random sample of 100 situations. Here, incorrectly aligned instances remain valuable for our data as negative samples in our later entailment task as they can be assigned a *No Relation* label.

**Social Norm Extraction.** We follow the formalism introduced by Forbes et al. (2020) in structuring descriptive norms as *rules-of-thumb*, or judgments of an action (i.e., "It's rude to run the blender at 5AM"). Taking 2 random top-aligned situations in our dataset, we manually annotate their corresponding Zhihu answer for social norms in rule-of-thumb form (RoT); an example of such an annotation is shown in Table 1. We then design a 2-shot prompt using these annotations (Table 3) and use GPT-3 (Brown et al., 2020) to extract novel rules-of-thumb from unseen answers. Running this model across all aligned situations and following a manual verification of faithfulness, we obtain a total of 6566 unique rules-of-thumb for our Chinese axis, relative to 1173 unique rules-of-thumb for our American axis sourced from SOCIALCHEMISTRY (note, here, that a rule-of-thumb may be associated with multiple situations).

**Identifying Cross-Cultural Norm Similarities & Differences as Textual Entailment with Explanations.** Under our formalism, a difference in cultural norms equates to a disagreement in judgments for an action relative to a given situation. Here, we structure the identification of social norm differences across cultures as an explainable textual entailment task (a.k.a natural language inference (NLI)) similar to e-SNLI (Camburu et al., 2018); a difference in cultural norms equates to a *contradiction* between norms of different cultures for a given situation, and vice versa for *entailment*. Given the recent success of human-AI collaboration frameworks (Wiegreffe et al., 2022; Bartolo et al., 2022; Chakrabarty et al., 2022), the complex nature of

*Given an input Text, respond with Social Norms that are assumed by the speaker of the text, in English. Social Norms are rules and standards that are understood by members of a group, and that guide or constrain social behaviors without the force of law. The Social Norms must be complete, correct, different, not contradicting with each other, inferable from text.*

**Text:** 关于尊敬长辈？因为传统。长辈始终是上辈。只要他不是有太过火大到让人无法原谅的错误。你就不能当面顶撞他。只能在人少的时候。试着和他理论。父母不光是站在长辈那里。更多是为了你。顶撞长辈。出发点就是一个错。让嘴巴大的人见了。你耳朵会红一辈子。养成这个习惯了。以后有困难了。也会更困难。
**Social Norms that we can infer from this text are:**
1. It is not polite to confront directly your elders even when elders are wrong
2. It is appropriate to argue with elders in private when you think they are wrong
3. It is shameful to point out their mistakes in front of crowds though elders are wrong

Table 3: Prompt used for the extraction of social norms in rule-of-thumb form from Zhihu questions and answers. The instruction precedes the in-context example; the original question (in red, equivalent to the situation) is followed directly by the answer, and sample annotated rules-of-thumb are listed in blue.

our task, and the need for interoperability in reasoning, we use Chain-of-Thought prompting (Wei et al., 2022) (see our prompt in Table 4) together with GPT-3 (`text-davinci-003`) to generate most of the data automatically before manual verification. In order to construct our data in the e-SNLI format and save on computation costs, for every given American norm associated with a given situation, we (1) *select* the most relevant Chinese norm for that situation as extracted in the previous step, (2) *identify* the inference relationship (entailment, contradiction, or no relation) between them, and (3) *generate* a free-text explanation for the identified relationship. When no Chinese norms are selected as relevant, a random norm is sampled from the aligned candidates, and an explanation is generated given the *No Relation* label. Using this framework, we generate social norm relationships and their justifications for all cross-culturally aligned situations in our dataset.

**Human Verification and Editing.** To ensure the quality and accuracy of our dataset of situated social norm relationships and their justifications across cultures, three annotators with significant (10+ years) lived experiences in both Chinese and American cultures further acted as annotators by jointly verifying and editing, if needed, the outputs of our framework (Figure 2). Specifically, authors ensured that (1) situations are correctly aligned (i.e., *entailment* or *contradiction* labels are assigned correctly to aligned norms, while *no relation labels* are assigned to misaligned norms), (2) norms generated by GPT-3 are not hallucinated, and (3) the explanations are correct. Annotators were able to discuss any questions and resolve any final disagreements with each other. In total, we find that 51% of norms had a *no relation* label, either due to misaligned situations or due to the absence of

extracted, relevant Chinese norms for corresponding US norms. Only a few instances (around 5%) had hallucinated norms; all were further corrected during annotation.

Generated explanations needed no further editing in 58.9% of the cases. In 38.1% of the cases, the inferred cross-cultural norm relation (and corresponding explanations) were deemed incorrect and required the revision of both the label and the explanation, while in the remaining instances (3.0%), generated explanations still required major revisions even though the relation label was deemed correct. Our final dataset contains 3,069 NLI instances of social norm relations across cultures with explanations. Out of these, 432 are *contradictions* (14%), 1059 are *entailments* (35%), and 1578 have a *no relation* label (51%). In total, aligned norms comprise 1173 unique rules-of-thumb from SOCIALCHEMISTRY (American axis) and 2273 as obtained from Zhihu using our framework (Chinese axis), with social norm description length averaging 63.5 characters. For social norm extraction, situation length was limited to 300 characters; examples from our dataset are shown in Table 2.

## 5 Experiments and Evaluation

**Task.** To test the ability of models to reason about social norms across cultures, we introduce the task of *explainable* social norm entailment. Our task is closely related to the explainable natural language inference task, e-SNLI (Camburu et al., 2018), where, in addition to the relation label, a model has to output a natural language explanation for its prediction as well. For every pair of US and Chinese norms of a cross-culturally aligned situation, we ask a model to predict the relationship between them (*Entailment*, *Contradiction*, or *No Relation*) and output an explanation for the relation. We test

Table 4: Chain-of-thought prompt used to identify (1) the most relevant (if any) Chinese norm given an American norm from an aligned situation, (2) determine the relationship between these norms, and (3) provide a free-text explanation justifying the inferred relation.

whether models fine-tuned on existing closely related and much larger textual entailment datasets (e.g., e-SNLI) and instruction-tuned models like FLAN-T5 can perform the task of explainable social norm entailment in addition to a simple fine-tuned baseline. In evaluation, we center our focus on explanation plausibility, testing if fine-tuning on our data may enhance a model's ability to reason about social norm entailment.

**Models.** We focus on smaller models on the order of 3B parameters and on non-OpenAI models, as an OpenAI model was used to generate our data. Here, we test the performance of a model fine-tuned on SOCIALCHEMISTRY rule elaborations (DREAM) and an instruction-tuned model (FLAN-T5) in few-shot settings, and further fine-tune joint self-rationalizing models on eSNLI and our dataset.

Prior work in interpretability (Wiegreffe et al., 2021) has shown that rationales from joint self-rationalizing models—predicting both the explanation alongside the relation label—are capable of producing faithful free-text rationales. Following prior work (Chakrabarty et al., 2022), we fine-tune a joint self-rationalizing T5 model in multiple settings, randomly splitting our data into 65% train, 10% validation, and 25% test set splits (Appendix Section C).

- **DREAM** (Gu et al., 2022a): an elaboration model that uses T5 to generate details about the input. Here, we test the DREAM-FLUTE (social-norm) variant (Gu et al., 2022b), which uses SOCIALCHEMISTRY rules-of-thumb as elaborations to provide an answer containing a label and an explanation. We evaluate this model in a 10-shot setting.

- **FLAN-T5-XL** (Chung et al., 2022): an enhanced version of the T5 (Raffel et al., 2020) 3B model fine-tuned on more than 1000 tasks, including e-SNLI (Camburu et al., 2018). We evaluate this model in a 10-shot setting.

- **T5-eSNLI**: a T5 3B model fine-tuned on the e-SNLI (Camburu et al., 2018) dataset. We fine-tune for two epochs with a batch size of 4096, with 268 total steps, and an AdamW optimizer with a learning rate of $5e-05$. We take the longest explanation per example in e-SNLI, as our data contains only one reference explanation. This leaves us with 549,367 training and 9,842 validation examples.

- **{mT5, T5}-SocNorm**: a T5 3B model fine-tuned on our Social Norm Explainable NLI dataset. We fine-tune for 20 epochs with a batch size of 256 (with 140 total steps) and an AdamW optimizer with a learning rate of $5e-05$. Note, here, that our training set of 2,000 examples (see Table 8) is *274 times smaller* than e-SNLI. We fine-tune a mT5 model in the same setting to explore how multilingual pretraining may impact in multicultural social norm understanding.

**Automatic Metrics.** Following Chakrabarty et al. (2022), we evaluate the quality of model-generated

| Model | F1@0 | F1@50 | F1@60 | %Δ(↓) |
|---|---|---|---|---|
| DREAM | 17.68 | 4.31 | 0.00 | 100.00 |
| FLAN-T5 | 25.85 | 8.92 | 7.11 | 72.50 |
| T5-eSNLI | 33.48 | 8.27 | 1.14 | 96.59 |
| mT5-SocNorm | 29.69 | 28.61 | 23.19 | 21.89 |
| T5-SocNorm | **54.52** | **51.68** | **43.07** | **21.00** |

Table 5: Automatic evaluation results of a series of competitive baseline models measured by F1 scores at three thresholds of the explanation score (0, 50, and 60, indicated as F1@0, F1@50, and F1@60, respectively), and models fine-tuned on our data. %Δ represents the percent decrease from F1@0 to F1@60.

explanations using the *explanation score* metric: an average of BLEURT (Sellam et al., 2020; Pu et al., 2021) and BERTScore (Zhang et al., 2019). We report the macro average F1 score at three thresholds of the explanation score: 0, 50, and 60. F1@0 is equivalent to simply computing the F1 score, while F1@50 counts only the correctly predicted labels that achieve an explanation score greater than 50 as correct.

As shown in Table 5, current off-the-shelf models under 3B parameters lack heavily in performance on our dataset, at most achieving an F1 score of 33.48, despite being fine-tuned on closely related tasks like e-SNLI. This verifies the need for a domain-specific dataset, as general NLI data is observed to be insufficient to perform well on our task. Furthermore, even the model fine-tuned on our data only achieves 54.52 F1 score, showing that there is still ample room for improvement in this domain.

When considering explanation quality, we see a very steep drop in performance from F1@0 to F1@60 for models fine-tuned on related tasks (72.50% for FLAN-T5 and 96.59% for T5-eSNLI), indicating that current datasets do not enable sufficient explanation quality for reasoning about social norm entailment. For the models fine-tuned on our data, the performance drop when accounting for explanation quality is not as sharp (≈ 21% for T5-SocNorm and mT5-SocNorm).

Interestingly, FLAN-T5 achieves a lower F1@0 score than a model fine-tuned on eSNLI, possibly because of interference from non-eSNLI-related tasks that it was fine-tuned on. Further investigating performance differences between T5-eSNLI and FLAN-T5 (see Table 6), we observe that FLAN-T5 struggles, in particular, to predict all relation classes equally well, instead predicting

| Model | Entail. | Contra. | NoRel. |
|---|---|---|---|
| FLAN-T5 | 1.36 | 13.56 | 62.62 |
| T5-eSNLI | 21.28 | 35.63 | 43.54 |
| T5-SocNorm | 56.6 | 47.0 | 59.97 |

Table 6: F1 score breakdown by relation label. FLAN-T5 mostly correctly predicts NoRelation class which allows it to achieve a higher F1@50 and F1@60 scores as these explanations are easier to generate. T5-SocNorm is more robust across relation classes.

most classes as *No Relation* (the majority class). This also explains the better performance of FLAN-T5 when accounting for the explanation score, as neutral explanations are easier to generate and typically more templatic in structure. T5-SocNorm and T5-eSNLI are seen as more robust in this regard.

**Explanation Quality.** Wiegreffe et al. (2021) introduced an automatic metric for textual explanation quality, termed as *rationale quality*, that compares the accuracy of the model with and without provided explanations. We fine-tune models to predict the relation label given an input and an explanation, in addition to giving only the input. When providing "gold" explanations, accuracy rises from 53.4% to 96.1% (with a rationale quality of 42.7), emphasizing the quality of textual explanations provided by our dataset.

**Human Evaluation of Generated Explanation Plausibility.** Three students with significant lived experiences (10+ years) in Chinese and American cultures assessed the quality of a subset of 50 randomly chosen model-generated explanations for correctly predicted labels, evaluating performance between the best performing model fine-tuned only on related tasks (T5-eSNLI) and the best-performing model that was directly fine-tuned on our dataset (T5-SocNorm). Each annotator was asked to rate which generated explanation they preferred, allowing for the presence of ties. Annotators showed an inter-annotator agreement of 62.4 in Fleiss' kappa (Fleiss, 1971), considered to be "substantial" agreement in existing literature. T5-SocNorm explanations are preferred for the vast majority of instances (73%); T5-eSNLI explanations were preferred in only 11% of instances, while explanations from both tied for the rest (16%). An example of a bad generation from T5-eSNLI is shown in Table 7; as shown, T5-SocNorm tries to determine the attitude the norms

expresses, while T5-eSNLI instead attempts to generate a rule-of-thumb (in red) instead. These results are indicative of how our data contains high-quality explanations—despite its small scale, models fine-tuned on it are able to produce better explanations compared to a model fine-tuned on the entirety of e-SNLI, which is 247 times larger in size.

| | |
|---|---|
| **T5-SocNorm** | The US norm expresses approval towards asking for money back, while the Chinese norm suggests that it is not polite to ask for it directly |
| **T5-eSNLI** | It is either OK to ask for your money back or not polite to directly ask |

Table 7: Example explanations generated by T5-SocNorm and T5-eSNLI for the situation in Figure 1. T5-eSNLI tries to generate a norm, while our model tries to explain the contradiction.

## 6 Cross-Cultural Norm Variation

Recalling that descriptive norms are situationally dependent, here, using our dataset, we test for factors driving these cross-cultural differences in social norms, testing specifically for (1) situational effects (i.e., *when* do norms differ?) and (2) descriptive effects (i.e., *what* do norms differ about?) across Chinese and American cultures.

**Situational Effects.** To capture thematic trends across situations, we train a 10-topic LDA model on preprocessed situations and manually label each topic with its most prominent theme; labeled situational topics alongside the top words that are associated with each are shown in Appendix Section D. Testing for the effect of topic-situational factors on differences in social norms, we measure the correlation strength between the probability of a situation belonging to a given topic against how likely norms will contradict for that given situation.

In this regression, we find that three situation topics positively predicted how likely norms would contradict between Chinese and American cultures for a given situation to a statistically significant degree. They are, as follows, *Lack of Intimacy/Separation* ($\rho = 0.07$, $p = 0.01$), *Family Discord/Divorce* ($\rho = 0.06$, $p = 0.01$), and *Loss of Family Connection/Changes in Life* ($\rho = 0.07$, $p = 0.02$).

Though these effect sizes remain small in scale likely due to the limited size of our dataset, these findings are consistent with contemporary studies

of Chinese cultural societies under the framework of social orientations (Yang, 1993). In Chinese culture, relational determinism strongly defines the social norms surrounding social interactions in everyday situations (Chen et al., 2013). One's relationship with another determines how one interacts with them; much distinction is placed within Chinese culture between one's own kith and kin (自己人, *zijiren*)—which mostly includes family members and familiar persons such as friends or classmates—and outsiders (外人, *wairen*), at a much stronger degree than that which is present in American cultures (Chen and Chen, 2004; Chua et al., 2009). Our results here show that in situations where this relationship distinction is apparent, or in cases of *change* in relationship type, the norms surrounding these situations are more likely to differ across Chinese and American cultures.

**Descriptive Effects.** As we have done for situational effects, here, we train a 10-topic LDA model on preprocessed norms in rules-of-thumb form, identify each norm topic's most prominent theme, and test for the effect of norm-topic factors on predicting differences in these norms between Chinese and American cultures. As before, labeled norm topics alongside their top words are shown in Appendix Section D.

Recalling that norms in rule-of-thumb form are judgments of actions, norm topics associate with *action* topics; a contradiction between cross-culturally aligned norms means that contrasting *judgments* are placed on those actions across Chinese and American cultures. Here, we find that two norm topics positively predicted how likely a contradiction would be present between the norms of our two cultures. They are, specifically, *Loss of Trust/Intimacy* ($\rho = 0.06$, $p = 0.05$) and *Support in Relationship* ($\rho = 0.04$, $p = 0.04$).

Interpreting these descriptive results in the context of relational determinism, our results above on situational effects, and through further qualitative analysis, our findings show that differences exist between Chinese and American cultures in judgments of actions that (1) would cause a loss of trust or intimacy and (2) are taken to support an individual close to you. These findings are consistent with longstanding work in Chinese familism (Ch'eng-K'Un, 1944; Yang, 1988; Campos et al., 2014), showing that exceptional accommodations in circumventing typical social rules are made, relative to American cultures, to pursue interpersonal and

especially familial harmony for those classed as one's own kith and kin (Dunfee and Warren, 2001; Chen et al., 2013).

# 7 Conclusion and Future Work

In this work, we sought to computationally model social norms across Chinese and American cultures, via a human-AI collaboration framework to extract and interpretably compare social norm differences in aligned situations across contexts. Our analyses here reveal several nuances in how social norms vary across these cultural contexts, incorporating principles of descriptive ethics that prior studies have often lacked thus far. We see our present work situated in the broader context of designing systems that are able to reason across languages and cultures in an increasingly interconnected global world. Here, we highlight a few directions we find exciting for future work; models, code, and anonymized data are made available for further research.[3]

Deviance from social norms can lead to miscommunication and pragmatic failure (Thomas, 1983). Integrating cross-cultural norm knowledge into *socially situated* communication systems (Hovy and Yang, 2021) could lead to fewer miscommunications in everyday situations, bridge cultures, and increase the accessibility of knowledge and of interpersonal technologies to a far greater extent than traditional translation technologies.

While our work has focused on measuring differences in social norms across cultures, culture, though important, is only one of many variables—like age and gender, among others—that affect descriptive ethical judgments (Kuntsche et al., 2021). Neither are norms fixed in time (Finnemore and Sikkink, 1998; Sethi and Somanathan, 1996); future work could, with the aid of social platforms like Zhihu and Reddit, quantitatively examine not only the evolution of social norms across cultures and platforms but also compare changes between cultures, further bridging theory and empirical study.

## Ethical Considerations

**Data Release and User Privacy.**   The study and data collection was approved by an Institutional Review Board prior to data collection. Nonetheless, while online data in platforms such as social media has opened the door to conducting computational

---

[3]https://github.com/asaakyan/SocNormNLI

social science research at scale, it is infeasible to obtain explicit consent for large-scale datasets such as ours (Buchanan, 2017). In order to preserve user privacy, we collected only publicly available data and analyzed it in aggregate, not at the individual level. Furthermore, we mirror former Twitter academic data release guidelines in that we release only the set of question and answer ids used in our analyses, which researchers are able to use together with the Zhihu API to obtain original discussion content.

## Limitations

**Cross-Cultural Demographic Differences.**   In studying cultural variations in social norms, we do not argue that culture as a variable alone explains the entirety of the observed variations. Most notably, *intra*-cultural variation in social norms also exists depending on the scale examined; for example, among multiple demographic groups or sub-communities (Plepi et al., 2022; Wan et al., 2023). Further, as the presence of variation in user and participant demographics undoubtedly remains between the data from our opposite cultural axes, it is crucial to interpret our results in the context of these differences and to take note that findings in our data are biased toward representing the viewpoints of individuals of these demographic groups.

**Value Pluralism.**   There are often more than one—and often contradictory—social norms that are relevant to any given social situation, which are often in tension with each other in many social dilemmas. By performing norm comparison using only the top-up-voted answer for given social situations, we necessarily limit the scope of our work to the social norms that are, by design assumptions and by proxy, the *most* accepted social norms for any given social situation. It is important to note that this does not preclude the possibility that similar social norms may remain valid for each culture at differing levels of acceptance. Here, while annotators with significant lived experiences in each culture sanity-checked that the entailment relations assigned between norms across cultures corresponded to actual cross-cultural differences and not simply because of value pluralism, this does not entirely ensure that the cross-cultural norm differences we observe are *not* because of value pluralism, as it is impossible for annotators to be aware of every single norm in an ever-evolving social landscape. We believe this to be a rich area of

future work to more quantitatively study instances of value pluralism even for norms of a single culture, and to see which norms ultimately "win over" the others for certain situations (Sorensen et al., 2023).

**Data Coverage.** It is unrealistic to expect that either our data or that of Social Chemistry can cover all possible social situations. As such, our findings only represent a subset of the true underlying cultural differences present between the norms of Chinese and American cultures. Furthermore, it seems intuitive that questions about non-controversial social situations "*everyone*" is familiar with will, if not be completely absent from online discourse, otherwise get lower representation and engagement. As we only extract from the most up-voted answer from each Zhihu discussion and treat it as the most broadly adopted norm as a simplifying assumption, an open question remains as to quantifying the "problem" of intra-cultural norm disagreements and to investigate genuine human variations in judgments of social norms (Plank, 2022). By releasing our data, we hope that future work can take into account less up-voted answers. In moving towards more representative studies, we encourage further work to examine these variations in intra-cultural norms and tease out the details of human behavior.

**Censorship and Moderation.** Chinese social media platforms like Weibo, and Wechat are mandated by law (Xu and Albert, 2014) to implement censorship policies on the content created by their users. Zhihu is no different; the presence of inherent content bias introduced by this form of active moderation of user content has an effect on influencing the landscape of public discourse (Yang, 2022b) and the data that we rely on to derive Chinese social norms. In particular, past work has shown the existence of censorship programs aimed at prohibiting "collective action by silencing comments that represent, reinforce, or spur social mobilization" across primary Chinese social media services (King et al., 2013). Comments that contain politically sensitive terms, such as mentions of the names of political leaders, are subject to a higher rate of deletions (Bamman et al., 2012). The extent to which these actions lead to a biased public representation of the norms and beliefs commonly shared in Chinese society remains unclear.

**Language Model Biases.** Advances in large language models like GPT-4 have allowed for greater possibilities in human-AI collaboration, as we have done here. Nonetheless, it is important to recognize that language models ultimately mimic patterns in their training data (Lucy and Bamman, 2021) and regurgitate structural and social biases (Bender et al., 2021). While we have incorporated these models under a human-AI collaboration framework where we externally verify, validate, and edit their output in certain cases to mitigate this risk, it would be remiss to say that we are capable of doing so entirely in lieu of effects such as priming influencing our decisions.

## Acknowledgements

We thank David Jurgens, Tuhin Chakrabarty, and the anonymous reviewers for their helpful comments, thoughts, and discussions. This research is being developed with funding from the Defense Advanced Research Projects Agency (DARPA) CCU Program No. HR001122C0034. The views, opinions and/or findings expressed are those of the authors and should not be interpreted as representing the official views or policies of the Department of Defense or the U.S. Government. Sky CH-Wang is supported by a National Science Foundation Graduate Research Fellowship under Grant No. DGE-2036197.

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

# Appendix

We provide as supplementary material additional information about our collected dataset from Zhihu, chosen hyperparameters for our models, topic model details from our analysis of cross-cultural norm variation, as well as the performance evaluations of larger models similar to the ones we used for data generation (GPT 3.5 and GPT-4) on our task.

## A Zhihu Statistics

Questions in Zhihu are tagged by users with topic categories, which serve as entities that individual users may browse, follow, and subscribe to. Figure 3 shows a breakdown of the top 20 user-tagged categories in our Zhihu questions dataset (508,681 unique questions) alongside their English translations, as well as a distribution of top-answer length. Following that of Social Chemistry, most topics here are related in content to relationships.

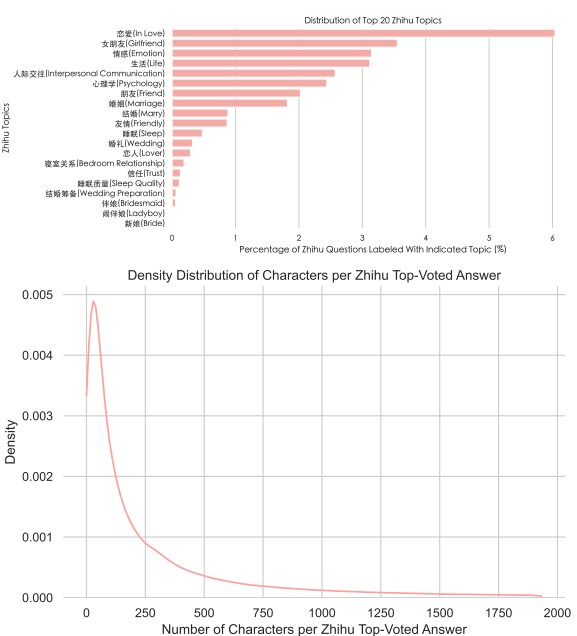

Figure 3: A top 20 user-labeled topics breakdown of Zhihu questions in our dataset (top) and the distribution of top-answer length measured in characters (bottom).

## B Hyperparameters

**Norm Extraction.** We use text-davinci-002 with the following hyperparameters:

```
temperature=0.7, max tokens=256,
top p=1, frequency penalty=0,
presence penalty=0.
```

**Inference Relation and Explanation Generation.** We use text-davinci-003 with the following hyperparameters:

```
temperature=0.7, max tokens=140,
top p=1, frequency penalty=0,
presence penalty=0.
```

**DREAM-FLUTE Instruction Modification.** We structure the input of social norm data into the DREAM-FLUTE instruction in the following manner:

*Premise: [Premise - social norm] US NORM. Hypothesis: [Hypothesis - social norm] CHINESE NORM. Is there a contradiction, entailment, or no relation between the premise and hypothesis?".*

We prefix the prompt with 10 examples from the validation set in the same format.

## C Dataset train test split statistics

|  | Train | Valid | Test |
|---|---|---|---|
| Contradiction | 279 (14%) | 39 (13%) | 114 (15%) |
| No Relation | 700 (35%) | 110 (37%) | 249 (32%) |
| Entailment | 1022 (51%) | 151 (50%) | 405 (53%) |

Table 8: Statistics of our cross-cultural Chinese-American social norm NLI dataset, stratified by entailment relation across train, validation, and test sets.

## D Topic Models

We train 10-topic LDA models using MALLET[4] to analyze cross-cultural social norm variations on both situational and descriptive effects, manually labeling each topic with its most prominent theme. Labeled topics and their top words are shown in Tables 9 (topics on situations) and 10 (topics on descriptive norms in rule-of-thumb form), with the topics that were statistically significant in the prediction of cross-cultural norm contradictions highlighted in **bold**.

---

[4]https://mimno.github.io/Mallet/topics.html

| Topic Theme | Top Tokens |
| --- | --- |
| 0. Self-care/ Emotional Turmoil | dog boyfriend living sad fears died wife doesn't speak everytime someone's petting hands washing morning teeth brush upset/irritated truth bed angrying stuff confess lie leads brother's attracted suffers agitated i'm |
| 1. School Bullying/ Toxic Relationship | school she's forced bullied pressuring friendship valentines absolutely engagement year disgusted people expecting fact bitchy remarry day tears burst criticism threw mother ditching mad |
| **2. Lack of Intimacy/ Separation** | crying boyfriend won't play stop can't reason don't consequences issues ill gravely starts panics lot unprofessional texting i've years that's reconciliation daughter's i've thing pathetic intimacy lacking public vulnerable separated |
| **3. Family Discord/ Divorce** | hate mother sister hating mom parents dad people wrong brother family father making wife cry starting divorce pissed what's resent girl kinda aunt brothers asked parent dislike grandmother marry genuinely |
| 4. Conflicts in Romantic Relationship | girlfriend boyfriend friend wanting mad friends telling birthday girl upset breaking break friend's hating family jealous relationship teacher boyfriend's hates angry boyfriends cheated gift dating depression wedding hang christmas likes |
| 5 Struggles in Marriage | birthday friends wanting fat dream husband night college part dont ashamed working cheat someone's frustrates affection reciprocate inability wife's complaining job open accepting remembered it's lies dropout couple boyfriend's pregnant |
| **6. Loss of Family Connection/ Changes in Life** | woman recently meet-up cancelled disappointed passing mother's blaming job dress original wear selling sick care taking form exist disappear fall dad wonders anymore wedding ex-bestfriend unfollow instagram despise |
| 7. Emotional Struggles/ Loneliness | fear son man trust thinking woman young married time covering welfare mom's unhappy constant lives longs toe camel bathtub facebook friending forgetting social co-worker...i'm knowing close lazy extremely dogs depressed |
| 8. Fear/ Uncertainty | love i'm don't boyfriend friend i'm married feel scared don't jealous anymore kids afraid mother life family yelling man younger cancer roommate dad advice can't doesn't telling terrified falling fiancée |
| 9. Unhealthy Relationship/ Mistreatment | mom ballet ago girls bad years girlfriend's ex-girlfriend scaring sounding insensitive ratting classmate hug holding she's doctor bring stressed teen friendship effort puts man's annoyed ex-wife learn lessons lie interested |

Table 9: Manually labeled *situation* topics and their top tokens, as captured from a 10-topic LDA model trained on cross-culturally aligned English Social Chemistry 101 situations.

| Topic Theme | Top Tokens |
| --- | --- |
| **0. Loss of Trust/ Intimacy** | abortion depressed hold sense loyal pet hatred talk kind doesn't hang past you're that's disgusted access hurtful brushing fiance's hide thoughts pushy depression pictures change breaking interact hard spite intimate |
| 1. Communication Issues/ Gifting | communication refuse girlfriend decision reach true public christmas dress lie grateful relationship lives steal assume enrolled stressful happily jeopardy admit you'll accepting interest interaction hangout debt loaned escalate friendly presents |
| **2. Support in Relationship** | it's expected people wrong family shouldn't friends good partner parents love hate rude bad significant understandable relationship normal feel don't expect things friend you're children feelings upset important talk friend's |
| 3. Social Media/ Social Bonds | families find you'll service wedding choose involved songs broken gifts media emotion cohesion foster behave society job online teaching ties sexuality interfere develop hurts schools professionalism poor health struggling kid's |
| 4. Responsibility/ Respect | put mother relationship long person's teacher understanding mental deserve lazy situation music partner aunt responsible wash face fix illegal material finances judge anger purposely concerns dog harass exchange college respond |
| 5. Care-taking/ Nurturing in Relationships | give relationship health frowned polite attention desires working inconsiderate single encouraged hang rest immoral random ignoring routine consistent lot provide assist won't strike arguments afraid information amends sister frequent concerns |
| 6. Personal Autonomy/ Control Over Life | relative fear calm death time members remember inappropriate methods elderly fights intimidate consent frustrate starting step betray relate reasons shy future beds stalk father business mind sense behavior control divorce |
| 7. Diversity/ Inclusivity | enjoy issues order college couples find aggressive place exclude petty fair gratitude trivial commitment grudges conversations you'll information attention based argue alive selfish lifestyle free live transgender nowadays socially dropout |
| 8. Trust/ Commitment | allowed spend common invite friend made plan supposed crush cheat interested activities hanging values chance contact tooth roommates toe lied betray deal extracurricular reasons member's finish students concerts effort honor |
| 9. Hardships/ Refusal | uncomfortable married accept frustrated interests bitter mom cheater hurtful who's therapy yous nasty taste stand lifetime bothering fears no-contact arguments phone purpose self-sufficient adults reaction hostile receiving health full shun |

Table 10: Manually labeled *descriptive norm* topics and their top tokens, as captured from a 10-topic LDA model trained on relevant rules-of-thumb from aligned cross-cultural situations.

# E Performance of GPT-3.5 and GPT-4

Here, we evaluate the performance of larger models similar to the one used to generate our data—GPT-3.5 and GPT-4. Both models achieve high F1@0 and F1@60 performance, as expected from their size and from the similarity of the data distribution to their outputs. Notably, however, the F1@60 score of GPT-4—the better performing of the two—remains lower in comparison with our best-fine-tuned model (41.18 vs. 43.07); furthermore, the relative decrease from F1@0 to F1@60 (%Δ) of GPT-4 remains higher than our model (34% vs 21%) as well, indicative of the possibility that distilled models may even surpass teacher models in explanation quality.

| Model | F1@0 | F1@50 | F1@60 | %Δ($\downarrow$) |
|---|---|---|---|---|
| GPT-3.5 | 52.21 | 47.30 | 30.41 | 41.75 |
| GPT-4 | 62.85 | 55.40 | 41.18 | 34.47 |

Table 11: F1 scores and percent decrease in F1 across explanation score thresholds for GPT-3.5 and GPT-4.