# OpenReview forum: "Sociocultural Norm Similarities and Differences via Situational Alignment and Explainable Textual Entailment"
_EMNLP/2023/Conference — EMNLP 2023 Main_

### Official Review · Reviewer_7iE3 · 2023-07-23

**Typos Grammar Style And Presentation Improvements:** N/A
**Soundness:** 4

**Excitement:**

3: Ambivalent: It has merits (e.g., it reports state-of-the-art results, the idea is nice), but there are key weaknesses (e.g., it describes incremental work), and it can significantly benefit from another round of revision. However, I won't object to accepting it if my co-reviewers champion it.

**Missing References:**

A recent paper that is relevant to yours. You are encouraged to cite this paper in your further version (not a reason to reject).
- SocialDial: A Benchmark for Socially-Aware Dialogue Systems (SIGIR'23)

**Paper Topic And Main Contributions:**

This paper proposed a human-AI collaboration framework for cross-culturaldescriptive norm discovery, as well as provided a new dataset for cross-cultural norms understanding with explainable textual entailment.

**Questions For The Authors:**

1. Do you have any plan to release this dataset  in further?

**Reasons To Accept:**

1. This paper is in well-written, easy and clear to follow.
2. This paper explored a task on cross-cultural social norm discovery between English and Chinese, two main languages around the world. The work would be meaningful and significantly contribute to the developmente of cross-cultural language development.
3. This paper proposed a Human-AI collaboration framework, which is easy to be extended for further work.
4. This paper provided supportive and comprehensive analysis and experimental to inspire how to ultize this datset in further.

**Reasons To Reject:**

1. Besides baseline models such as T5, a simple tailored model should be introduced to help formulate the task of cross-cultural norm discovery and expainable textual entailment.
2. Details about this dataset should be introduced. e.g., how to split the 3069 social norms in train/dev/test? what's average length of each social norm and dialogue context?

**Reproducibility:**

4: Could mostly reproduce the results, but there may be some variation because of sample variance or minor variations in their interpretation of the protocol or method.

**Reviewer Confidence:**

5: Positive that my evaluation is correct. I read the paper very carefully and I am very familiar with related work.

---

> ### Author Rebuttal · Authors · 2023-08-28
>
> We thank the reviewer for their comments and appreciation of our contributions to the study of cross-cultural social norms. We also thank the reviewer for pointing out a very relevant work and will cite it in our paper.
>
> * Reason to reject 1: We would like to emphasize 5 modeling strategies investigated in this study: model fine-tuned for elaborations on SocialChemistry (DREAM), state-of-the-art instruction-tuned model (FLAN T5), model fine-tuned on a large amount of data for a related task (T5 e-SNLI) and the model fine-tuned on the dataset. In addition, we conducted an experiment with fine-tuning a multilingual mT5 model (see response to Reviewer 3gbm), as well as few-shot performance of GPT-4 and GPT-3.5 (see response to Reviewer nK7w) and plan to include these results in the paper, adding 3 more modeling approaches to our investigation. We are not sure what is meant by a simple tailored model, perhaps that is satisfied with the model fine-tuned on the dataset (T5-SocNorm)?
>
> * Reason to reject 2:  Please refer to Appendix C for dataset split. The average length of the social norms is 63.5 characters. For social norm extraction, situation length was limited to 300 characters. We will provide more detailed statistics in the appendix.
>
> * Question 1: Yes, the dataset will be publicly released.

---

### Official Review · Reviewer_nK7w · 2023-08-02

**Soundness:** 4

**Excitement:**

3: Ambivalent: It has merits (e.g., it reports state-of-the-art results, the idea is nice), but there are key weaknesses (e.g., it describes incremental work), and it can significantly benefit from another round of revision. However, I won't object to accepting it if my co-reviewers champion it.

**Paper Topic And Main Contributions:**

This paper studies the social norm relations (similarities & differences) between the U.S. and Chinese by proposing a novel corpus. Specifically, this paper collects the contexts from the existing English corpus (Social-Chem-101) and Chinese QA platform (Zhihu). In order to investigate the cross-cultural social norm, the authors put these bi-lingual pairs into the same situations and use LLMs to annotate the entailment relations along with explanations. After careful human verification and evaluation, the authors show the high quality and challenge of this dataset, where some widely-adopted LMs fail at solving this task.

**Questions For The Authors:**

Same as the "Reasons to Reject".

**Reasons To Accept:**

1. **An interesting research problem**. To my knowledge, this might be the first paper that addresses the cross-cultural social norm identification task. The authors also conduct experiments to show the challenge for current LMs to cope with this problem, implying the emergency of future studies.
2. **A high-quality corpus**. This paper proposes a corpus for this task accordingly. By adopting a human-AI collaboration pipeline, the dataset covers more than 3k social situations, the corresponding bi-cultural rule-of-thumb, and explainable relation tags. The thorough analysis also exhibits statistical alignment with existing literature on human societies.
3. **Wrtting**. This paper is well-formed and is easy-to-follow.

**Reasons To Reject:**

1. **Limited contributions**. The authors only test the capacity of some well-known LMs; there are limited insights or investigations on the modeling strategies of this problem. For example, it should be useful to investigate the reason for such bad F1@0 performance. Personally, I think this entailment task is not that difficult; more related analysis should help a lot for future research. To be honest, I didn't learn much after reading this paper.
2. **Some other LLMs**. The authors used GPT-3 (text-davinci-003) to generate the rule-of-thumb, however, I am curious about why they didn't try with other LLMs with more power capacity, such as ChatGPT. Similarly, I think it is also necessary to test whether the vanilla LLMs (e.g., GPT-4) can efficiently deal with this problem.

**Reproducibility:**

4: Could mostly reproduce the results, but there may be some variation because of sample variance or minor variations in their interpretation of the protocol or method.

**Reviewer Confidence:**

3: Pretty sure, but there's a chance I missed something. Although I have a good feel for this area in general, I did not carefully check the paper's details, e.g., the math, experimental design, or novelty.

---

> ### Author Rebuttal · Authors · 2023-08-28
>
> We thank the reviewer for their comments and appreciation of novelty of our contributions.
>
> * Reason to reject 1: We would like to emphasize 5 modeling strategies investigated in this study: model fine-tuned for elaborations on SocialChemistry, state-of-the-art instruction-tuned model (FLAN T5), model fine-tuned on a large amount of data for a related task (T5 e-SNLI) and the model fine-tuned on the dataset. All of these approaches result in a relatively weak performance which serves as objective evidence of the inherent difficulty of the task (even though personally it may feel easy). Furthermore, the focus of our work is *explainable entailment*, the performance of which we try to measure with F1@50 and F1@60, *which is an even more difficult task.* We agree with the reviewer that more investigations could be explored in future work, and hope that our release of the dataset and the initial investigations of 5 different approaches to this task will be a reasonable first step for future work.
>     * In addition, we conducted an experiment with fine-tuning a multilingual mT5 model (see response to Reviewer 3gbm), as well as few-shot performance of GPT-4 and GPT-3.5 (see below) and plan to include these results in the paper, adding 3 more modeling approaches to our investigation.
>
> * Reason to reject 2: We found that the capacity of text-davinci-003 was enough, in conjunction with meticulous human validation described in Section 4, to generate a high-quality dataset for our task. Due to the time-consuming nature of human validation, certain state-of-the-art models may advance in the meantime. However, we encourage future work to consider state-of-the-art models in our rapidly evolving field.
>     * We also implemented an experiment from Table 5 with GPT-4 in a 10-shot setting (same as for instruction-tuned FLAN-T5 baseline) and found the performance to be 62.85, 55.4, 41.18 for F1@0, F1@50, F1@60 respectively. Note that despite being a much more powerful model (order of magnitude higher number of parameters and reinforcement learning training), it performs worse than the fine-tuned 3B T5-SocNorm both in terms of absolute score for F1@60 (41.18 vs. 43.07) and relative decrease (34% vs. 21%). In a similar experiment with GPT-3.5, the results are 52.21, 47.3, 30.41, respectively, supporting a similar conclusion. We would be happy to report these results if provided additional space in camera-ready. That being said, we note that the scope of our work was focused on smaller, more efficient models under 3B parameters (as defined in abstract, see line 025).

---

### Official Review · Reviewer_3gbm · 2023-08-02

**Soundness:** 3

**Excitement:**

3: Ambivalent: It has merits (e.g., it reports state-of-the-art results, the idea is nice), but there are key weaknesses (e.g., it describes incremental work), and it can significantly benefit from another round of revision. However, I won't object to accepting it if my co-reviewers champion it.

**Paper Topic And Main Contributions:**

This paper presents an interesting exploration into how NLP systems can handle socio-cultural differences in social acceptability of situations. Given the social chemistry dataset (Forbes et al, 2020) of situation-RoT tuples, authors scrape a Chinese advice-seeking forum (Zhihu) to create a parallel corpus of US-centric and Chinese-centric situations. They then generate/obtain RoTs for both situations, and manually label whether the RoTs disagree or agree, to find contradictory/different acceptability judgments, along with an explanation of why. The paper then describes modeling experiments on the pairwise task of predicting agreement/disagreement between social norms for a given situation, showing that models benefit substantially from being finetuned on the task (as opposed to on related tasks like e-SNLI). Authors conclude with a small qualitative investigation of why situations and norms are different.

**Questions For The Authors:**

See first reason to reject

**Update post rebuttal** Changing my soundness score from 2->3

**Reasons To Accept:**

- More work like this one is needed to mitigate the huge US-centrism bias in NLP research
- The introduced dataset is useful to understand cultural differences in social acceptability in social situations
- I particularly appreciated the analysis in §6 of why situations and judgments might be different in different cultures, especially since it was well-grounded in socio-cultural studies.

**Reasons To Reject:**

- My main concern about this paper is that it seemingly glosses over the fact that there are more than one---often contradictory--- social norms that are relevant to a situation, which could often support opposing judgments for the situation. For example, for the situation "*staying home instead of going to work because you're sick*" evokes both norms that supports the situation ("*it's good to take care of your health*") and that oppose it ("*you should go to work*"). Such a discordance is just evidence that often social norms are in tension with one-another (value pluralism). My question is, how many of the contradictions in the final dataset are actually contradictions due to cross-cultural differences, vs. contradictions that are just due relevant social norms in US-centric situations being sampled and ranked for relevance differently than in Chinese-centric situations, and this really more due to value pluralism that is inherent to many social dilemmas?
- The choices of modelling experiments is reasonable but does not yield many insights that would be useful for other researchers interested in adapting NLP to other cultures. Authors could, for example:
  - Run some experiments or ablations to compare whether multi-lingual pretrained models have better performance at predicting different social acceptability judgments / predicting the NLI-style task compared to monolingual models
  - Explore whether training specifically on social chemistry would help with the task.
- Nitpick: though I appreciated the analyses in §6 greatly, the effect sizes of the correlations were very very small (likely due to the size of the dataset, especially given how little contradictions there were). Perhaps authors could hedge their claims more in this part to indicate these small effect sizes?

**Reproducibility:**

3: Could reproduce the results with some difficulty. The settings of parameters are underspecified or subjectively determined; the training/evaluation data are not widely available.

**Reviewer Confidence:**

4: Quite sure. I tried to check the important points carefully. It's unlikely, though conceivable, that I missed something that should affect my ratings.

**Typos Grammar Style And Presentation Improvements:**

- Tables 1 and 2 are slightly wider than the text columns.
- Nitpick: I would have appreciated if more details about the various annotations / verifications done were included in the main body of the paper instead of the appendix

---

> ### Author Rebuttal · Authors · 2023-08-28
>
> We thank the reviewer for their engagement with our work and insightful comments.
>
> * Reason to reject 1: Absolutely, value pluralism is one limitation for doing alignment in this fashion—On this point, during the data validation stage described in Section 4, we had annotators with significant lived experiences (10+ years) in both cultures sanity check that the entailment relations assigned between norms across cultures corresponded to actual cross-cultural differences and not simply because of value pluralism.  While this does not entirely ensure that the cross-cultural norms we observe are not because of value pluralism (as it’s not possible for annotators to be aware of every single norm), this was a good mitigation step and acted as a good validation check for our method that is based on a human-AI collaboration framework. We thank the reviewer for pointing out this omission and will clarify the details of this in the camera-ready, and include an acknowledgement of value pluralism as a potential limitation in the Limitations section. We believe this to be a rich area of future work to more quantitatively study instances of value pluralism even for norms of a single culture, and to see which norms ultimately “win over” the others for certain situations.
>
> * Reason to reject 2: We thank the reviewer for their suggestions for additional experiments.
>     - Our framework and modeling experiments can be easily adapted for other cultures since they only take the basic NLI set-up as assumptions, and our framework for data generation is also easily extendable to other cultures.
>     - *We fine-tuned a multilingual mT5 model on our dataset to explore how multilingual pre-training aids in social norm understanding.* We found that the fine-tuned mT5-XL model achieves 29.69, 28.61, and 23.19 scores for F1@0, F1@50, and F1@60 respectively. While the absolute scores are lower than the monolingual T5, the relative decrease from @0 to @60 is in line with the T5 model at 22%). We will report this and perhaps other ablations in the paper given the additional page for camera-ready. We also implemented experiments with GPT-3.5 and GPT-4 (see response to reviewer nK7w).
>     - We also would like to note that one of the baselines (DREAM) was fine-tuned on SocialChemistry (lines 372-374, 386-394), but that alone did not seem to yield significant improvements. We may include more experiments with socialChemistry given the additional page for camera-ready.
>
> * Reason to reject 3: We thank the reviewers for pointing this out, and yes—we will adjust the hedging language of our claims to reflect this effect size.

---

### Meta-Review · Area_Chair_i6z1 · 2023-09-11

**Recommendation:** 4

**Metareview:**

This paper contributes a new task and dataset: comparing social norms across Chinese and American cultures. They build upon the Social Chemistry dataset to present a novel corpus of 3k social situations in the Chinese language and analyze cross-cultural norm differences.

Several reviewers commented on the importance of this paper as the “first paper that addresses the cross-cultural social norm identification task” and “mitigat[ing] the huge US-centrism bias in NLP research”.

The author response was very thorough, causing at least one reviewer to increase their soundness score. Although I see this paper as primarily a task and dataset paper, the authors have also conducted many empirical experiments that show there is still room for models to improve on this task and dataset. For reviewers 7iE3 and nK7w reasons to reject as “some other LLMs” and “a simple tailored model”, these arguments seem to fall under the “The authors could also do [extra experiment X]” heuristic/shortcut discouraged in the reviewer guidelines: https://2023.aclweb.org/blog/review-acl23/#2-check-for-lazy-thinking

---

### Decision · Program_Chairs · 2023-10-07

**Decision:**

Accept-Main

**Comment:**

This paper contributes a new task and dataset: comparing social norms across Chinese and American cultures. They build upon the Social Chemistry dataset to present a novel corpus of 3k social situations in the Chinese language and analyze cross-cultural norm differences.

Several reviewers commented on the importance of this paper as the “first paper that addresses the cross-cultural social norm identification task” and “mitigat[ing] the huge US-centrism bias in NLP research”.

The author response was very thorough, causing at least one reviewer to increase their soundness score. Although I see this paper as primarily a task and dataset paper, the authors have also conducted many empirical experiments that show there is still room for models to improve on this task and dataset. For reviewers 7iE3 and nK7w reasons to reject as “some other LLMs” and “a simple tailored model”, these arguments seem to fall under the “The authors could also do [extra experiment X]” heuristic/shortcut discouraged in the reviewer guidelines: https://2023.aclweb.org/blog/review-acl23/#2-check-for-lazy-thinking